# A Dynamic Procedure to Detect Maximum Voluntary Contractions in Low Back

**DOI:** 10.3390/s23114999

**Published:** 2023-05-23

**Authors:** Xun Wang, Karla Beltran Martinez, Ali Golabchi, Mahdi Tavakoli, Hossein Rouhani

**Affiliations:** 1Department of Mechanical Engineering, University of Alberta, Edmonton, AB T6G 2R3, Canada; 2Department of Civil and Environmental Engineering, University of Alberta, Edmonton, AB T6G 2R3, Canada; 3EWI Works International Inc., Edmonton, AB T6E 3N8, Canada; 4Department of Electrical and Computer Engineering, University of Alberta, Edmonton, AB T6G 2R3, Canada

**Keywords:** surface electromyography, low back muscles, trunk bending, ergonomic risk, MVC

## Abstract

Surface electromyography (sEMG) is generally used to measure muscles’ activity. The sEMG signal can be affected using several factors and vary among individuals and even measurement trials. Thus, to consistently evaluate data among individuals and trials, the maximum voluntary contraction (MVC) value is usually calculated and used to normalize sEMG signals. However, the sEMG amplitude collected from low back muscles can be frequently larger than that found when conventional MVC measurement procedures are used. To address this limitation, in this study, we proposed a new dynamic MVC measurement procedure for low back muscles. Inspired by weightlifting, we designed a detailed dynamic MVC procedure, and then collected data from 10 able-bodied participants and compared their performances using several conventional MVC procedures by normalizing the sEMG amplitude for the same test. The sEMG amplitude normalized by our dynamic MVC procedure showed a much lower value than those obtained using other procedures (Wilcoxon signed-rank test, with *p* < 0.05), indicating that the sEMG collected during dynamic MVC procedure had a larger amplitude than those of conventional MVC procedures. Therefore, our proposed dynamic MVC obtained sEMG amplitudes closer to its physiological maximum value and is thus more capable of normalizing the sEMG amplitude for low back muscles.

## 1. Introduction

Surface electromyography (sEMG) is an important technique in ergonomic risk assessments and the diagnosis of musculoskeletal disorder (MSD) [1]. sEMG data can be interpreted based on the amplitude and frequency features of the collected sEMG signals. However, the sEMG amplitude can vary among individuals and different measurement trials due to several factors, such as the electrode placement location, the thickness of soft tissue between the muscle and electrode, and skin preparation [2]. Therefore, to consistently characterize muscle activity based on the sEMG signal amplitude among individuals and trials, the raw sEMG signal is usually normalized to that of the maximum voluntary contraction (MVC). The sEMG amplitude expressed as a percentage of MVC amplitude can then be used to evaluate muscle fatigue, the risk of MSD, and diagnose medical conditions [3].

For different muscles, different MVC exercises are used to record the sEMG amplitude when only a target muscle is activated under a proper external force to reach its maximum contraction. This procedure is straightforward for major muscles of the upper and lower limbs. Yet, due to the complex musculature of the low back and the presence of several layers of muscles under the skin, several muscles contribute to trunk bending and resistance against an external force. Therefore, it is hardly possible to: (i) isolate the contraction of a muscle while maximizing its contraction to resist an external force, and (ii) ensure that the same group of muscles are involved in the MVC task and the movement during the actual test [4]. As a result, sEMG electrodes do not necessarily record the maximum activity of a targeted muscle during the MVC procedure [5,6]. Currently, there are two ways to normalize sEMG data: (i) using MVC data, and (ii) performing maximum amount of activity during the actual test procedure [7,8]. In the first approach, the MVC for back muscles (erector spinae and latissimus dorsi [9,10]) is collected during maximum back extension against a manual force on a flat plane [11]. This MVC procedure only concentrates on individual muscles and only on a small range of motion for the lumbar spine joint. Thus, the sEMG amplitude during this MVC may be smaller compared to that generated during the performance of actual tasks [2]. On the other hand, when performing a maximum amount of activity during a physical task, it is common to not obtain consistent results due to high inter-trial variability [12]. 

In general, the sEMG amplitude recorded for low back muscles, namely the left/right latissimus dorsi and left/right thoracolumbar fascia, which are commonly used during physical tasks [4,13], may be greater during the actual task measurement compared to that generated during conventional MVC procedures for these muscles. This is because the low back muscles’ recruitment and synergy vary depending on the specific task performed. Therefore, to effectively normalize the sEMG amplitude for the aforementioned low back muscles, this study aimed to propose novel dynamic MVC exercise procedures based on an actual task; in this case, this was a trunk bending task. We did not aim to instruct the participant to recruit specific muscles, but it was assumed that by performing the same actual task, the participant would use the same muscles, and thus, we could obtain the highest possible sEMG amplitudes in the dynamic MVC procedure (by exerting a maximum force) for the muscles involved in the actual test; in this case, this was a trunk bending task. Additionally, the validity of our proposed procedure was experimentally investigated, and its results were compared with those collected via a conventional MVC exercises procedure. 

## 2. Materials and Methods

### 2.1. Dynamic MVC Measurement Procedure

Usually, static exercises are used for MVC collection for low back muscles; some common examples are seen in Figure 1: in Figure 1a, the participant needs to hold their torso in the air while an external force is applied on their back, in Figure 1b the participant needs to bend their legs 90 degrees, and pull them up, in Figure 1c the exercise involves pulling legs and arms as high as they can, and lastly in Figure 1d the participant stands close to a wall with an external load on their upper back, and they need to push back. However, these methods do not always record a muscle activity signal higher than that obtained during an actual task. To illustrate this, we recorded the sEMG amplitude from the right latissimus dorsi during one of the most common MVC procedures (Figure 1c) and compared it to that obtained during a typical material handling task. The results are shown in Figure 2, where it can be observed that the collected sEMG amplitude while the participants performed the task was larger than the sEMG amplitude during MVC collection. 

Thus, we expanded upon the MVC procedures to include dynamic tasks to explore the highest sEMG amplitude recorded from the involved low back muscles. During the dynamic MVC procedure, participants were instructed to lift a 45 lbs. weight only using their low back muscles, which meant that they needed to keep their elbows and knees straight during the movement (Figure 3). Additionally, an external force was applied to control the pace and make sure the muscles experienced maximum contraction. A metronome working at 40 bpm was used, and participants were instructed to finish each motion within each beat. The external force varied among participants based on their body strength to produce maximum muscle contraction and make the participant follow a fix pace produced by a 40 bpm metronome during the MVC procedure. Participants were instructed to perform the lifting task three times as slowly and smoothly as possible. Notably, low back muscle recruitment, and thus, the recorded crosstalk may have varied from one dynamic task to another and between static and dynamic tasks. Therefore, the purpose of our proposed method was not to eliminate crosstalk produced by other muscles or exert control over which muscle group was activated during the MVC, but rather to normalize the recorded sEMG amplitudes during the actual trial to a reference from the same trial with maximum load to achieve the same muscle recruitment.

### 2.2. Experimental Procedure

To investigate the efficiency of the dynamic MVC measurement procedure, a set of experimental procedures was designed. Data were collected from 10 able-bodied participants without any history of musculoskeletal disorders (6 males and 4 females; body mass: 61.2 ± 8.7 kg; body height: 171.2 ± 48 cm; age: 23.8 ± 1.5 y.o.). Four EMG sensors (Trigno Avanti EMG sensor, Delsys, Natick, MA, USA) were placed on both the right and left latissimus dorsi and thoracolumbar fascia, the muscles mainly involved in weightlifting tasks [14]. The location of sensors is shown in Figure 4, and they were placed according to the recommendation in [15]. To measure MVC, we used the four conventional procedures seen in Figure 1 (trunk bending, leg bending, trunk-leg combined, and standing posture, which are commonly used and recommended by authors of previous studies [5,6,7,8]) and our novel dynamic MVC technique.

Finally, the participants were instructed to perform a manual lifting task three times at their preferred pace. The task consisted of lifting a 45 lbs. weight from the floor to the participant’s chest, and then lowering it to the floor again. sEMG data collected during the performance of four conventional MVC exercises and via the proposed dynamic MVC procedure were normalized. Then, the normalized sEMG results for each muscle were compared together.

### 2.3. Data Processing

#### 2.3.1. EMG Processing

The raw sEMG signal’s amplitude can have a range of ±5000 µV, with its energy being concentrated mostly between 20 Hz and 150 Hz. The sEMG recording is usually rectified and band-pass filtered before further data interpretation [16,17,18]. In this study, data were collected at the sampling frequency of 2148.15 Hz (using EMGworks Acquisition software, Delsys, Natick, MA, USA). The sEMG signal was processed as follows:Remove the baseline error using the medium value during a quiet, lying down period.Band-pass filter the EMG signal using a 4th-order Butterworth filter, with cut-off frequencies of 10 Hz and 500 Hz.Perform full wave rectification.Smooth the results using a moving average filter with 500 sample points.Calculate the root mean square (RMS) of the sEMG amplitude during the working period.Normalize the results of five different MVC procedures.

#### 2.3.2. Statistical Test

We hypothesized that the best MVC collection procedure would obtain the highest sEMG amplitude, and thus, the lowest normalized sEMG amplitudes in the same actual lifting test. To investigate if those undergoing the dynamic MVC procedure outperformed those in undergoing the other conventional MVC measurement procedures, we used the paired Wilcoxon signed-rank test (since the distribution of the data was not normal). We compared the results normalized according to the dynamic MVC measurement procedure to those normalized according to each of the conventional MVC procedures. The significance level (dynamic MVC procedure versus each of the conventional MVC procedures) was set to 0.05 [19,20].

## 3. Results

The sEMG amplitudes during the lifting task, which were normalized according to all four conventional MVC procedures, were frequently larger than 100% (Table 1 and Figure 5). Only the dynamic MVC procedure was used to obtain normalized sEMG amplitudes of less than 100% in the majority of the participants. Table 1 demonstrates the high variability of the obtained normalized sEMG amplitudes among participants, since the low back muscles can be differently activated among individuals, even when they are performing the same task (due to subtle differences in task execution strategies or different sport or work routines). Therefore, the paired tests were performed on an individual basis, and this high variability did not impact the statistical results. The dynamic MVC procedure was used to obtain the lowest normalized sEMG amplitudes among these five methods for all four target muscles, with almost all *p* values being less than 0.05, according to Table 2.

According to Figure 5 and Table 2, we observed significant differences between the sEMG results obtained using conventional MVC procedures and the dynamic MVC procedure, except for trunk bending MVC and standing MVC from the right latissimus dorsi muscle. Meanwhile, for both sides of the thoracolumbar facia muscle, the muscle activity data normalized according to the dynamic MVC procedure were significantly lower than those of each of the four conventional MVC procedures.

## 4. Discussion

The normalization of sEMG amplitude translates raw voltage data into muscle activity relative to MVC and can provide meaningful information about muscle fatigue and MSDs [21,22]. Due to the complex musculature of the low back, conventional MVC measurement procedures usually fail to obtain sEMG that are larger than those collected during any weightlifting task, which is a condition that is assumed for all MVC collections. This study aimed to address this challenge by defining a novel, dynamic MVC procedure for low trunk muscles that obtained the largest sEMG amplitudes during MVC collection and a normalized sEMG amplitude during a weightlifting task, always with a value of less than 100%. We also experimentally compared the performances of the proposed MVC procedure with those of the conventional MVC procedures proposed in the literature.

We observed that in the dynamic MVC procedure, we obtained higher sEMG amplitudes than we did during each of the four conventional MVC procedures, and the sEMG amplitudes normalized according to the dynamic MVC procedure were, in general, lower than those normalized according to conventional MVC procedures for the four studied low back muscles (all *p* values for Wilcoxon signed-rank test were smaller than 0.05, except for trunk bending and standing MVC procedure for right latissimus dorsi). Additionally, we observed that the sEMG amplitudes normalized according to the dynamic MVC procedure were from 18% to 46% lower than those normalized according to conventional MVC procedures. 

Low back muscles are prone to the crosstalk effect between multi-muscles, which could be a reason for obtaining a higher sEMG amplitude during a desired task than that obtained during the MVC procedure. Another reason could be due to the misplacing of sEMG electrodes [23,24]. For body parts, such as the low back, and during tasks, such as trunk bending, in which the EMG recording will be affected by several muscles [25], introducing an MVC procedure that isolates the contraction of a single muscle could be challenging, and thus, it may be reasonable to use dynamic MVC procedures instead of static procedures. This dynamic MVC procedure should involve motions similar to the actual trunk bending task, but with maximum voluntary muscle contraction, and thus, it is unlike conventional MVC procedures. Our proposed dynamic MVC procedure did not intend to eliminate the impact of crosstalk and electrode misplacement in the MVC data, but we intended to obtain similar levels of crosstalk and errors during the MVC procedure and the actual task. Nevertheless, further investigations are warranted to assess their effectiveness in mitigating various errors in sEMG recording in future studies. Although we did not discuss it here, we also implemented the dynamic MVC procedure during the in-field weightlifting test and observed its feasibility and efficiency. 

This study exclusively focuses on creating an MVC procedure for the low back muscles during a forward-bending task. However, the effectiveness of the proposed dynamic MVC concept method should be explored in future studies for (i) different tasks involving the low back muscles, and (ii) other body parts with complex musculature. Additionally, the feasibility of the proposed MVC approach should be further investigated in a larger and more diverse participant population.

## 5. Conclusions

This study introduced a novel, dynamic MVC measurement procedure for low back muscles. The dynamic MVC measurement procedure showed a better performance compared to that of the conventional static MVC tests for low back muscles, which is evident from us having obtained higher sEMG amplitudes during the dynamic MVC procedure compared to those obtained using the conventional MVC procedures. The proposed dynamics MVC procedure was also the only MVC procedure that obtained normalized sEMG amplitudes, which were always less than 100%, unlike the conventional MCV procedures. The efficiency of our proposed dynamic MVC should be further investigated for other skeletal muscles.

## Figures and Tables

**Figure 1 sensors-23-04999-f001:**
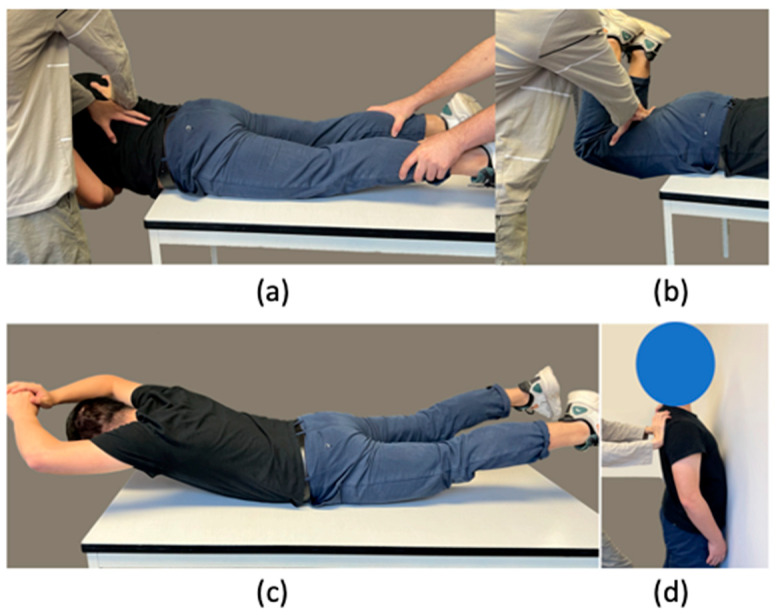
Four conventional MVC measurement exercises, trunk bending (**a**), leg bending (**b**), leg-trunk combined (**c**), and standing (**d**); this is similar to [4].

**Figure 2 sensors-23-04999-f002:**
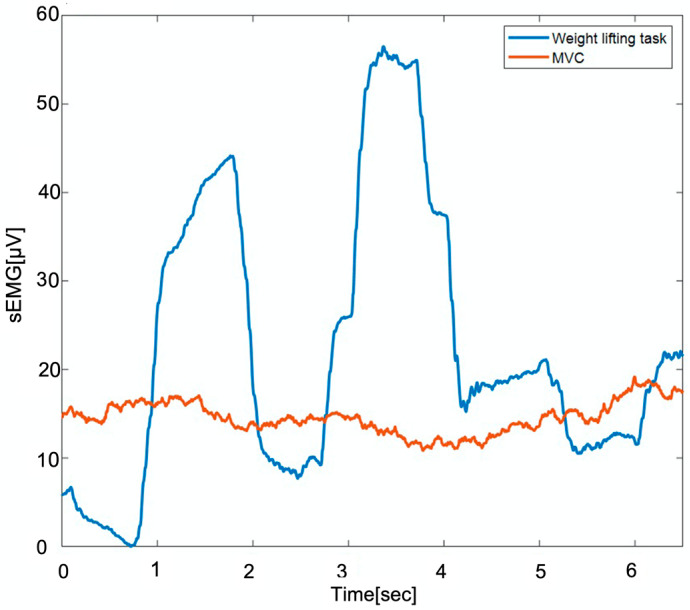
sEMG signal amplitude collected from right latissimus dorsi during a weightlifting test and MVC collection.

**Figure 3 sensors-23-04999-f003:**
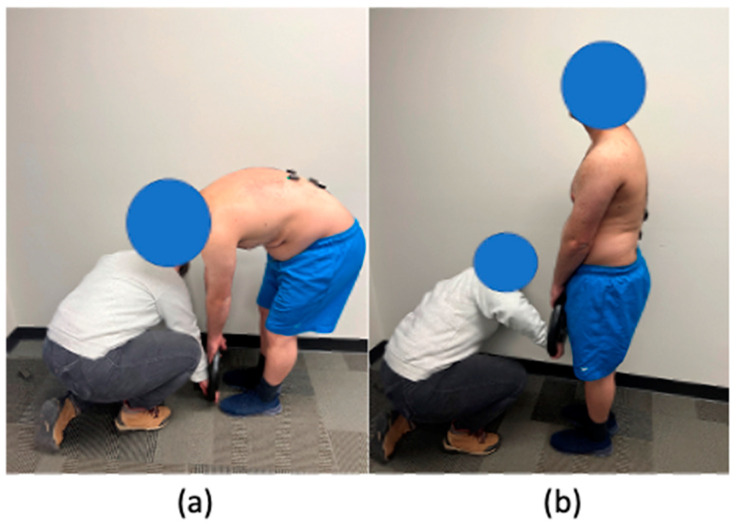
Dynamic MVC procedure start posture (**a**) and end posture (**b**).

**Figure 4 sensors-23-04999-f004:**
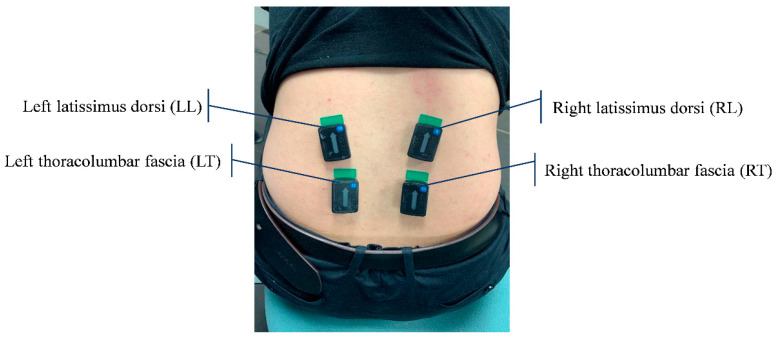
Placement of EMG sensor modules on low back muscles.

**Figure 5 sensors-23-04999-f005:**
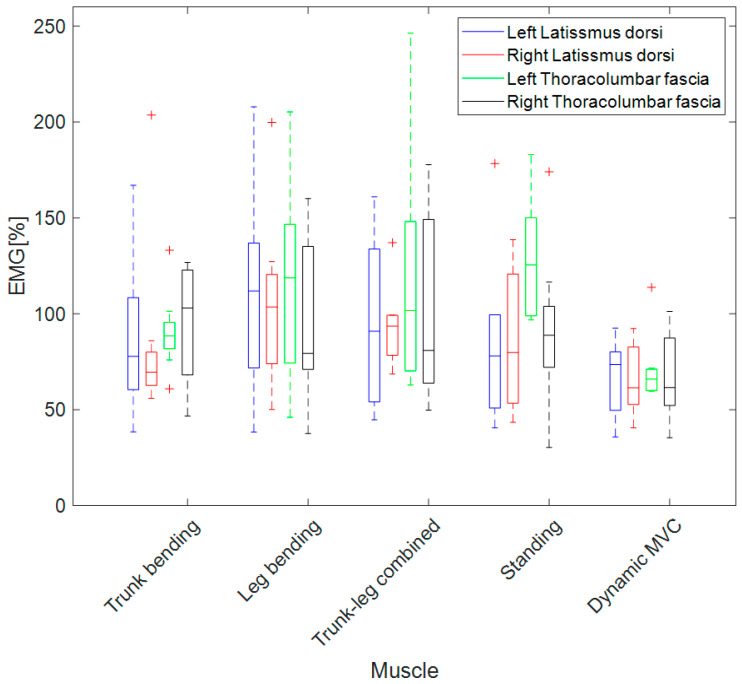
sEMG amplitude for the trunk bending task normalized according to four conventional MVC procedures and dynamic MVC procedure, expressed as boxplots among participants. “+” indicates the outliers in the boxplot.

**Table 1 sensors-23-04999-t001:** Mean value and standard deviation (SD: among participants) of normalized data expressed in percentage.

	MVC Test	Trunk Bending	Leg Bending	Leg-Trunk Combined	Standing	Dynamic
Muscles		Mean Value/%				
**Left Latissimus dorsi**		112.35	110.94	116.51	84.44	66.87
**Right Latissimus dorsi**		85.48	105.74	93.48	86.23	65.81
**Left Thoracolumbar fascia**		90.70	116.31	118.64	128.63	70.95
**Right Thoracolumbar fascia**		95.16	96.08	101.92	91.74	67.33
		**SD/%**				
**Left Latissimus dorsi**		48.82	51.01	38.35	42.15	19.38
**Right Latissimus dorsi**		45.47	42.86	20.01	34.08	17.10
**Left Thoracolumbar fascia**		19.50	48.47	58.00	30.21	16.80
**Right Thoracolumbar fascia**		30.57	41.21	47.93	38.78	21.81

**Table 2 sensors-23-04999-t002:** *p* Values for paired Wilcoxon signed-rank test performed on normalized EMG data for duplicated task; EMG data normalized according to conventional MVC procedures were compared with those normalized according to dynamic MVC. *p* Values smaller than 0.05 are shown with bold font and asterisks.

	MVC Test	Trunk Bending	Leg Bending	Trunk Leg Combined	Standing
Muscles					
**Left Latissimus dorsi**		0.0391 *	0.0234 *	0.0391 *	0.0234 *
**Right Latissimus dorsi**		0.1094	0.0234 *	0.0156 *	0.0547
**Left Thoracolumbar fascia**		0.0078 *	0.0156 *	0.0391 *	0.0078 *
**Right Thoracolumbar fascia**		0.0078 *	0.0156 *	0.0078 *	0.0391 *

## Data Availability

The data presented in this study are available on request from the corresponding author. The data are not publicly available due to privacy.

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
