# Peer review of "A Dynamic Procedure to Detect Maximum Voluntary Contractions in Low Back"

_sensors, 2023, doi:10.3390/s23114999_

Round 1

Reviewer 1 Report

The authors reported a detailed dynamic MVC procedure inspired by weight-lifting task. With this method, the normalized sEMG amplitude showed a much lower value than by conventional MVC procedures. This is a potentially interesting approach, but the concept of this work is defective. I am not in the favor of its publication based on the following points:

1.       Line 44&Line 102: The author mentioned that the sEMG collection was not accurate during the MVC task because several muscles contribute to the trunk bending and resistance against an external force. However, in this work, EMG sensors were still used for detecting the sEMG signal experiment, which meant that the data in this experiment still be affected by the crosstalk of surrounding muscles. The reviewer expresses confusion about this experimental design.

2.       Line 45&line 85: The author mentioned that the limitations of existing methods for collecting the sEMG of lower back area is that the participant is hard to drive the same group of muscles involved in the MVC task and the movement during the test. In new dynamic procedure, reviewer is very confused about how to allow the participant to control specific muscle groups during weight lifting movements.

3.       Figure 5: In this box plot, 9 boxes were not marked with dotted lines, and replaced with the unexplained "+" and "-" signs, which affected the understanding of the work. The reviewer suggests that the author carefully check the figures, or make a reasonable explanation for special signs.

4.       Table 1: The SD of the data is too large, some even higher than 50%, which reduced the credibility of the conclusions. The reviewer recommends expanding the sample size in subsequent tests.

5.       Line 190: In the comprehensive results and discussion section, the author had proven that the dynamic MVC strategy proposed in this article was different from conventional MVC procedures, but the article had not explained the potential of the new method to overcome errors caused by crosstalk and electrode misalignment through persuasive data. The reviewer believes that some convincing data should be added to this section to support the conclusions.

6.       In common statistical descriptions, the letter P should be in italic font, and 5% should be changed to 0.05.

However, I encourage author to resubmit this work after revision. If the author add some data and innovative descriptions, including any further work recommended, it may then be suitable for publication.

Reviewer 2 Report

Dear authors

This was an interesting study about EMG testing for low back muscles; below my suggestions:

- better clarify inclusion criteria to undergo EMG testing: able bodied participant is an unclear label; were they screened before the test? some conditions could affect testing, i.e. sagittal plane asymmetries, hyperlordosis,  sport, work, ecc

- better clarify the technique to place EMG sensors (points of reference, initial position); again, specify the text 

- results section: figure five needs a comment on EMG amplitude comparison

Reviewer 3 Report

Thank you for allowing me to review this interesting study. Overall, the study has raised a very interesting point of discussion. I believe that this study has provided novel findings in this area, allowing readers to think more deeply about what is happening around this issue.

First of all, I would like to share the need to carry out works like the one you present. They are necessary for the advancement of science in the field they study. The objective of the manuscript is clear and consistent. The study has been an interesting reading, it is necessary to know the reality of the sector on which the work emphasizes.

Likewise, reasons are highlighted that justify the importance in a broad context and the current state of the subject investigated. The study is clearly defined and indicates the intention and meaning of the work. The objective to be tested in the study is recorded. The text is understandable and makes clear the main objective of the work and the main conclusions.

Author Response

We thank the reviewer for their kind words. We have improved the following sections based on your recommendation: cited references, research design and methods description, clarity on the results, and conclusion supported on the results.